# Brain Activity in the Prefrontal Cortex during Cognitive Tasks and Dual Tasks in Community-Dwelling Elderly People with Pre-Frailty: A Pilot Study for Early Detection of Cognitive Decline

**DOI:** 10.3390/healthcare9101250

**Published:** 2021-09-23

**Authors:** Kohei Maruya, Tomoyuki Arai, Hiroaki Fujita

**Affiliations:** Department of Physical Therapy, Faculty of Health & Medical Care, Saitama Medical University, Moroyama-Town, Saitama 350-0496, Japan; arai_tm@saitama-med.ac.jp (T.A.); fuhi55@saitama-med.ac.jp (H.F.)

**Keywords:** brain activity, functional near-infrared spectroscopy, fNIRS, pre-frailty, prefrontal cortex

## Abstract

We aimed to detect brain abnormalities during cognitive and motor tasks in older individuals with pre-frailty, as this could prevent dementia. Sixty elderly participants (mean age: 76.3 years; 27 healthy and 33 with pre-frailty) were included, and their motor function, cognitive function, and dual-task abilities (gait with calculation and while carrying a ball) were evaluated. Total hemoglobin (t-Hb) was measured using functional near-infrared spectroscopy (fNIRS) during tasks and resting periods. The pre-frailty group had a slightly lower gait speed than the healthy group, but there was no significant difference in cognitive function. In the pre-frailty group, the t-Hb values during the normal gait and cognitive tasks were higher than the resting value in the right prefrontal cortex, while in the healthy group, only the word frequency task (WFT) was higher. Furthermore, in the WFT, the t-Hb values were significantly lower in the pre-frailty group than in the healthy group. The results showed that pre-frail subjects had lower brain activation during the WFT in the right prefrontal cortex and excessive activity during walking, even without a noticeable cognitive decline. The differences in cerebral blood flow under the pre-frailty conditions may be a clue to detecting cognitive decline earlier.

## 1. Introduction

Around the world, there were 9.9 million new cases of dementia in 2015, with one new case every 3 s. Furthermore, dementia has a significant economic impact and affects many social issues, making prevention important [1]. Mild cognitive impairment (MCI) is considered to be a preclinical stage of dementia. Patients with MCI are more prone to dementia than individuals with normal cognitive function. However, even MCI can improve to normal cognitive function with appropriate intervention [2]. The criteria for MCI proposed by Petersen et al. [3] includes a person or a family member complaining of cognitive decline and objective impairment of one or more cognitive functions. However, activities of daily living are normal and not characteristic of dementia. Thus, MCI is difficult to detect early because, as patients do not have a significant cognitive decline, they can continue to perform normal daily activities.

It has been reported that MCI is related to motor dysfunction [4,5], and motor dysfunction has been reported to progress more rapidly following the development of MCI than in the preceding years [6]. Community-dwelling adults, those with slower walking speeds, and those with a greater decline in walking speed over time are at greater risk of developing dementia, independently of changes in cognition [7].

Frailty is an example of impaired motor function and is associated with aging. This condition is a common geriatric syndrome that increases the risk of catastrophic health and function declines in elderly people [8]. The diagnostic standard is based on the Cardiovascular Health Study (CHS) criteria suggested by Fried et al. [9]. There are five CHS criteria: weight loss, weakness, exhaustion, slowness, and low activity. This frailty model is a phenotypic model that focuses on the vulnerability of physical functions. Frailty is also associated with cognitive decline and risk of dementia [10]. In other words, a state of frailty and decline in motor function are thought to be associated with a decline in cognitive function. However, the mechanism underlying this association remains unclear. In particular, there are no reports of pre-frailty and brain activity. 

Near-infrared spectroscopy (NIRS) is a non-invasive adjunct diagnostic tool for measuring brain function using near-infrared light irradiation at the body surface to detect changes in the concentration of oxidized hemoglobin during a task, enabling real-time examination of changes in blood flow in the brain and evaluation of functional status. Patients with dementia or MCI have been reported to exhibit abnormalities in cerebral hemodynamic and oxygenation parameters compared with healthy individuals [11,12]. Differences between healthy young adults and healthy older adults in terms of brain activity have been reported, with older adults showing a pattern of reduced brain activity compared with young adults [13,14]. However, the differences in brain activity between healthy elderly people and those with motor function decline are unclear. An analysis of these differences may reveal the mechanisms underlying motor and cognitive decline. Therefore, the present study aimed to address this gap in knowledge and provide evidence that may contribute to the prevention of cognitive decline. In particular, detection of brain abnormalities at the pre-frailty stage (which involves a less pronounced loss of motor function than frailty) could enable intervention at an earlier age and the subsequent prevention of dementia. We compared the brain activity of pre-frail and healthy elderly participants during cognitive, motor, and dual tasks, as measured by functional near-infrared spectroscopy (fNIRS).

## 2. Materials and Methods

### 2.1. Ethical Considerations

The study was prepared in accordance with the “Ethical Guidelines for Medical Research Involving Human Subjects” and was approved by the ethics board of the Faculty of Health and Medical Care of Saitama Medical University (No. 194). 

We gave a written and verbal explanation of the study to community-dwelling elderly people aged 65 years or older, who were gathered by the Community General Support Center and the city, and obtained their written consent.

### 2.2. Participants

We recruited community-dwelling elderly volunteers in the Saitama Prefecture, Japan, aged over 65 years. Each of the participants independently joined a local health prevention program in the community. All participants agreed to undergo the procedures and consented to the use of their data for research. 

### 2.3. Measurements

Cognitive function was measured using a shortened version of the color Stroop test (shortened Stroop test) and the Rapid Dementia Screening Test for the Japanese population (RDST-J). Both tests can be performed in less than 5 min, and we adopted the two tests in consideration of the time constraints and burden on the participants. The shortened Stroop test was performed according to the method described by Hosoda et al. [15], consisting of 20 slides, each with the name of a color (“red,” “blue,” “green,” or “yellow”) printed in a different color. These were displayed on a slide show, and the participants were asked to name the ink color instead of the word written. The examiner switched to the next slide only after the participant had answered correctly. There was no limit of attempts or time to complete the test. Before the test, the participants were given a practice round, and the time taken to answer the 20 slides correctly was then measured using a stopwatch. The RDST-J consists of a word frequency task (WFT) and a numeral mutual conversion task. This test has been reported and validated as an effective screening test for dementia and mild cognitive impairment [16]. In the WFT, participants were asked to record as many supermarket and convenience store goods as they could think of in 1 min. Participants were awarded 8 points for recalling 14 or more words, 6 points for recalling 11–13 words, 4 points for 8–10 words, 2 points for 5–7 words, and 0 points for recalling 4 words or less. In the numeral mutual conversion task, participants had to convert Arabic numerals to Chinese numerals and Chinese numerals to Arabic numerals, and they received 1 point per each correct answer. Each task was measured once, and the scores obtained in the WFT and RDST-J tests were used for analysis. The Stroop test and WFT have been reported to be associated with frontal lobe function (executive and attention functioning), primarily involving the dorsolateral prefrontal cortex [17,18].

Physical performance was evaluated based on hand grip strength (HGS), the duration of single-leg standing with eyes open, normal gait speed (NGS), maximum gait speed (MGS), dual-task gait, and two-step test. HGS was measured using a Smedley grip strength meter (T.K.K.5001, Takei Scientific Instruments Co., Ltd., Niigata, Japan). Participants gripped the instrument as hard as they could once with each hand, and the maximum value of the two sides was used for analysis. Single-leg standing was measured for a maximum of 120 s. In this test, participants were asked to stand on their left or right leg for about 5 s before the measurement was taken and were then asked to decide on which leg they could stand on better/easier. The two-step test was conducted according to a previously published method [19]. Participants were asked to take two consecutive steps as far forward as possible and to stop at the second step. The test was repeated if the participants lost balance during the task or if they did not stop after the second step. In addition, participants were asked not to jump. The test was performed until two valid results were obtained. The maximum value was selected and divided by the participant’s height to provide a two-step value. The NGS and MGS were measured using a stopwatch while participants walked on a 5-m walkway. Dual-task gait analysis assumed two patterns: gait with calculation and walking while carrying a ball. For gait with calculation, participants were asked to walk along a 5-m walkway while continually subtracting 3 from 50 (e.g., 50, 47, 44, 41, 38, etc.). For walking while carrying a ball, the participants were asked to carry a 6-cm diameter plastic ball balanced on a teaspoon in their dominant hand while walking as quickly as possible a distance of 5 m without dropping the ball. The time to completion was measured for both tasks. If they failed, they had to start over from the beginning, but even for those participants who failed multiple times, it was possible to achieve success after three iterations. The best time for completing the task was used in the analysis. Calculating gait requires working memory, executive function, and attention. Carrying a ball gait requires executive function and attention, which are mediated by the prefrontal cortex.

Participants were asked to complete a questionnaire to evaluate whether they had lost 2–3 kg of weight in the previous 6 months and if the previous 2 weeks had been unusually tiring. We also collected information on years of education and frequency of exercise (walking or other exercise) per week, and the Japan Science and Technology Agency Index of Competence to Assess Functional Capacity (JST-IC) was determined for each participant. The JST-IC was created to measure higher-level competence according to Lawton’s hierarchical model of competence and to measure the competencies required for older individuals living alone to be independent and lead an active daily life within the living environment of modern active older citizens [20]. The JST-IC consists of four categories: use of new technology, information gathering, management of living, and social participation. Each category has four yes/no questions. The total score ranges from 0 to 16 points, with a higher score indicating higher competence. 

Brain activity was measured using a wearable two-channel continuous-wave NIRS (HOT-2000; NeU Co., Ltd., Tokyo, Japan). The device consists of a light source and two light detectors, which are placed at a distance of 1.0 cm and 3.0 cm from the source. The light source was a single LED (800 nm) for the measurement of total hemoglobin (t-Hb). It estimates the concentration change of t-Hb using the modified Beer-Lambert law, and the two source-detector (SD) pairs enable the dual SD regression method. The 1.0 cm detector captures the shallow blood-flow signal, which is dominated by systemic components in the superficial layers of the scalp. The deep signal from the 3.0 cm detector contains both the superficial and cortical blood-flow components. The dual SD regression method formulates a deep signal as a linear combination of shallow and neural signals. The reliability of the device has been reported previously [21]. Each SD pair was placed on the forehead above the eyebrows (Figure 1), and light intensity was sampled at 10 Hz. The data were transferred via Bluetooth to a tablet device. For statistical analysis, we used the average values at rest and during tasks, which were detrended and filtered using a low pass filter (0.05 Hz) for the removal of cumulative effects and artifacts caused by minor movements of the participant.

### 2.4. Classification of Pre-Frailty

Participants were classified using the criteria of the Japanese version of the Cardiovascular Health Study (J-CHS) [22], which consists of five items: weight loss (weight loss of 2–3 kg over the past 6 months), low muscle power (HGS: male < 26 kg, female < 18 kg), exhaustion, slow gait speed (NGS < 1.0 m/s), and low physical activity. Pre-frailty was defined as meeting one or two of the five criteria, and frailty was defined as meeting three or more. However, there were only three participants with frailty and were therefore excluded from the analysis. Participants who met none of the criteria were classified as healthy.

### 2.5. Statistical Analysis

Continuous variables were compared between the healthy and pre-frailty groups using an independent t-test, and categorical variables were compared using the chi-squared test. In addition, fNIRS data were analyzed using two-way repeated measures analysis of variance (two-way ANOVA). Statistical analyses were conducted using JMP ver. 13.0 for Mac (SAS Institute Japan, Ltd., Tokyo, Japan). 

## 3. Results

Sixty-five participants were enrolled in this study. However, five participants were excluded due to previous surgery for brain tumors (1), missing data (1), and frailty (3); hence, 60 participants were included in the final analysis. The participants’ characteristics are presented in Table 1. The average age was 76.3 years, and the average number of years of education was 11.8 years. About 80% of the respondents were women. The prevalence of pre-frailty was 55%. The two-step value, NGS, MGS, and gait speed calculations were found to be significantly lower in the pre-frailty group than in the healthy group. Age, sex, years of education, cognitive function, and JST-IC scores were not significantly different between the groups.

In the left prefrontal cortex, the t-Hb value increased significantly when the WFT in both groups was not significant (Table 2). Meanwhile, in the right prefrontal cortex, t-Hb in the healthy group was significantly increased in the WFT compared with the resting period (−0.064 ± 0.297 mMmm to 0.203 ± 0.224 mMmm, *p*< 0.01). In the pre-frailty group, t-Hb values were increased in normal gait, the shortened version of the Stroop test, and WFT compared with the resting values. 

On the right side, the pre-frailty group had lower scores on the WFT than the healthy group (0.071 ± 0.100 mMmm vs. 0.203 ± 0.224 mMmm, *p* = 0.004). Interaction effects were not significant on either side in any task, although in the right side, the normal and calculating gait had a tendency to increase more in the pre-frailty group than in the healthy group (F = 3.365, *p* = 0.072; F = 3.762, *p* = 0.057).

## 4. Discussion

Although MCI is different from dementia, this condition has been reported to be associated with abnormal cerebral blood flow (an oxidative parameter). Furthermore, MCI is related to motor dysfunction, which may increase the risk of cognitive decline in elderly individuals. The present study enrolled participants who were living independently in the community to evaluate differences in brain activity between participants with pre-frailty and healthy elderly individuals. 

We found that the pre-frailty group did not exhibit a decline in cognitive function, although the two-step value, NGS, and MGS values were lower than those in the healthy elderly group. The cut-off score of the RDST-J in the cognitive screening was 7 points [16]. Therefore, neither the healthy group nor the pre-frailty group was classified as having cognitive impairment (i.e., MCI or mild dementia), as their scores were above this cut-off. 

In the present study, we found that the bilateral prefrontal cortex was activated during the WFT. Logan et al. [23] reported that young adults exhibit significant activation in the left prefrontal cortex during intentional word tasks, although they found no difference between left and right activation in elderly subjects. This phenomenon is known as the hemispheric asymmetry reduction in older adults (HAROLD phenomenon) [24] and is thought to be caused by the use of contralateral brain regions that compensate for declining abilities in the elderly [25]. We believe that these results support the validity of the present study, which was based on data obtained using a simple and portable fNIRS instrument. 

We found that activation of the right prefrontal cortex during the WFT was significantly lower in the pre-frailty group than in the healthy group. This reduced activation in the right prefrontal cortex of pre-fail individuals is presumably due to a reduced ability to functionally complement. Holtzer et al. [26] and Beurskens et al. [27] reported that older adults exhibit reduced activation in the prefrontal cortex during different tasks compared with younger subjects. Furthermore, decreased physical activity has been reported to cause reduced blood flow to the brain [28], leading to frailty [29]. In addition, a previous report found that older adults who go outdoors on a daily basis had greater oxygenated hemoglobin changes in the inferior frontal gyrus compared with those who do not have this habit [30]. In other words, it is suggested that decreased physical activity and motor function lead to a decrease in brain blood flow and function, which in turn reduces complementary functions. With aging, cognitive functions are expected to decline. The J-CHS criteria used in this study considered the level of physical activity. Therefore, elderly participants who were categorized into the pre-frailty group may have suffered a decline in motor function as a result of decreased physical activity, leading to pre-frailty. In the pre-frailty group, the results of fNIRS during normal gait and in the Stroop test were significantly increased compared with the resting period, which was not seen in the healthy group. The interaction effects of normal and calculating gait in the pre-frailty group tended to increase more than in the healthy group. The reason for this might be that the patients in the pre-frailty group needed to increase their brain activity because their walking and brain functions were reduced compared with the healthy group.

The findings of this study demonstrate that elderly individuals with pre-frailty have low activation during the WFT, even when there is no noticeable decrease in cognitive or motor functions. In addition, the results indicated that pre-frail subjects required more brain activity during walking because of their reduced walking ability. Although elderly people with impaired physical function often have impaired cognitive function, the differences in cerebral blood flow changes displayed in those with pre-frailty may be a precursor to impaired cognitive function. Furthermore, our findings indicate that changes in cerebral blood flow occur before the development of frailty or MCI, highlighting a potential approach for early intervention for MCI and dementia.

This study has some methodological limitations that should be acknowledged. As this was a cross-sectional study, the causality of the relationship between the reduction in brain activity and pre-frailty remains unknown. Another potential limitation is that some risk factors for pre-frailty or abnormal brain activity could not be assessed. As the total number of participants was only 60, it is unclear whether this was representative of the entire elderly population, and selection bias cannot be ruled out. In addition, our use of two-channel NIRS did not enable the detailed measurement of other brain areas. In addition, due to the nature of the device, only the t-Hb value was displayed, while the detailed measurements of oxidized hemoglobin and deoxidized hemoglobin values were not obtained. In addition, it has been reported that the rate of increase in prefrontal oxygenation decreases with age, resulting in cognitive decline [31], but in the present study, the rate of increase could not be examined because the mean value between tasks was used. However, one of the strengths of this study is that, although the number of participants and effect size were small, we were able to identify differences in brain activity changes between the groups even among those with pre-frailty who only had a slight decline in motor function. In the future, larger and longitudinal studies, including participants with reduced motor function are warranted to confirm our findings.

## Figures and Tables

**Figure 1 healthcare-09-01250-f001:**
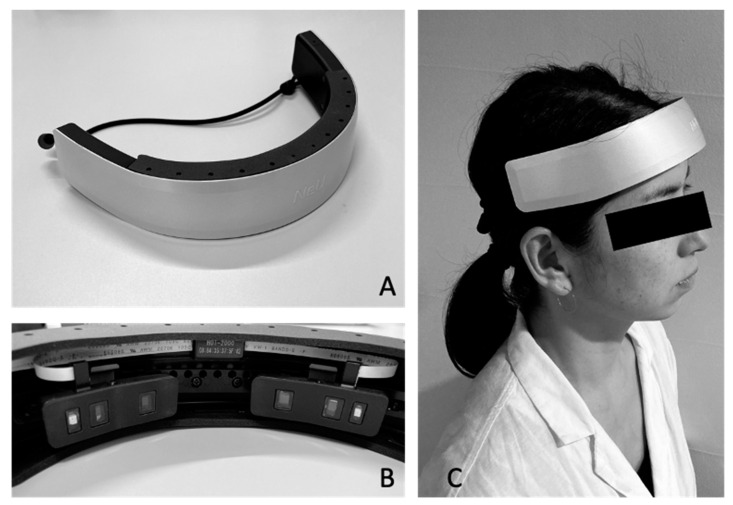
Photographs of the device used to measure brain activity (HOT-2000). (**A**) overall device view. (**B**) Close-up view of the sensor units. (**C**) Photograph of a subject wearing the device.

**Table 1 healthcare-09-01250-t001:** Baseline characteristics and motor and cognitive performance of participants.

	Overall N = 60	Healthy N = 27 (45%)	Pre-Frailty N = 33 (55%)	*p*-Value
Age, years	76.3 ± 5.5	76.5 ± 4.8	76.1 ± 6.2	0.785
Female, number (%) †	47 (78.3)	19 (70.4)	28 (84.9)	0.176
BMI, kg/m^2^	22.3 ± 3.0	22.1 ± 2.9	22.5 ± 3.0	0.636
Educational years	11.8 ± 2.0	11.8 ± 2.1	11.8 ± 2.0	0.951
HGS, kg	22.6 ± 5.7	24.1 ± 5.2	21.3 ± 5.9	0.062
SLS, s	48.2 ± 42.9	59.5 ± 46.6	38.9 ± 37.8	0.063
Two-step value, m/m	1.37 ± 0.16	1.42 ± 0.13	1.33 ± 0.17	0.035
NGS, m/s	1.20 ± 0.21	1.29 ± 0.20	1.12 ± 0.18	0.002
MGS, m/s	1.48 ± 0.19	1.55 ± 0.18	1.43 ± 0.19	0.011
Shortened Stroop test time, s	30.8 ± 7.7	30.8 ± 8.2	30.9 ± 7.5	0.973
Calculating gait speed, m/s	0.86 ± 0.32	0.96 ± 0.31	0.77 ± 0.31	0.026
Carrying a ball gait speed, m/s	1.03 ± 0.29	1.06 ± 0.28	1.00 ± 0.30	0.413
WFT words, number	12.7 ± 3.3	12.7 ± 2.9	12.7 ± 3.6	0.966
RDST-J total score	9.5 ± 2.3	9.7 ± 2.2	9.4 ± 2.5	0.691
JST-IC total score	11.8 ± 2.6	12.2 ± 2.0	11.5 ± 3.0	0.286

Data are presented as mean ± standard deviation. *p*-values were calculated using an independent *t*-test or the chi-squared test (indicated with †). Abbreviations: BMI, body mass index; HGS, hand grip strength; SLS, single-leg standing; NGS, normal gait speed; MGS, maximum gait speed; WFT, word frequency test; RDST-J, Rapid Dementia Screening Test; JST-IC, Japan Science and Technology Agency Index of Competence to Assess Functional Capacity.

**Table 2 healthcare-09-01250-t002:** Comparison of fNIRS results in the left and right prefrontal cortex.

*Left*	Healthy		Pre-Frailty		Between *p* Value	Interaction F-Value	Interaction *p*-Value
Rest	−0.117 ± 0.301		−0.079 ± 0.263		0.597	−	−
Normal gait	−0.094 ± 0.242		0.012 ± 0.256		0.108	0.814	0.371
Calculating gait	−0.123 ± 0.274		−0.041 ± 0.195		0.181	0.219	0.642
Carrying a ball gait	−0.043 ± 0.302		−0.042 ± 0.158		0.988	0.162	0.689
Shortened Stroop test	−0.016 ± 0.173		−0.006 ± 0.173		0.828	0.195	0.660
WFT	0.153 ± 0.200	**	0.123 ± 0.121	**	0.484	0.610	0.438
** *Right* **	**Healthy**		**Pre-Frailty**		**Between** ***p* Value**	**Interaction** **F-Value**	**Interaction** ***p*-Value**
Rest	−0.064 ± 0.297		−0.150 ± 0.233		0.215	−	−
Normal gait	−0.046 ± 0.232		0.017 ± 0.173	**	0.233	3.365	0.072
Calculating gait	−0.128 ± 0.213		−0.051 ± 0.161		0.119	3.762	0.057
Carrying a ball gait	−0.028 ± 0.229		−0.212 ± 1.095		0.393	0.221	0.640
Shortened Stroop test	−0.024 ± 0.162		0.039 ± 0.272	*	0.306	2.384	0.128
WFT	0.203 ± 0.224	**	0.071 ± 0.100	**	0.004	0.292	0.591

Data are presented as mean ± standard deviation. Units of activation are mMmm. *: *p* < 0.05, **: *p* < 0.01 vs. rest by paired *t*-test. Comparison of *p* values by t-test. *p*-value by two-way repeated measures analysis of variance. Abbreviations: fNIRS: functional near-infrared spectroscopy; WFT, word frequency test.

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
