# Peer review of "Brain Activity in the Prefrontal Cortex during Cognitive Tasks and Dual Tasks in Community-Dwelling Elderly People with Pre-Frailty: A Pilot Study for Early Detection of Cognitive Decline"

_healthcare, 2021, doi:10.3390/healthcare9101250_

Round 1

Reviewer 1 Report

We read with interest the article by Maruya et al assessing "brain activity in the prefrontal cortex during cognitive tasks in elderly people with prefrailty: which involved hemoglobin (t-Hb) evaluation using functional near-infrared spectroscopy (fNIRS) during tasks and resting periods. The data presented are modest and in terms showing a lower brain activation during the WFT test in the pre-frail subjects had lower brain activation in the right prefrontal cortex with no noticeable cognitive decline; these results of blood flow were suggested to be used as biomarkers for cognitive decline. the last statement is an overstatement as the study had a low number of participants with only 60 subjects and this would make this study a Pilot Study which I would suggest that the Title change for this pilot study as there is no power analysis for the low number of participants.

A major concern is the lack of IRB approval and participant consent for the study which needs to be presented or else the article needs to be rejected.

It would be interesting if the authors could perform  some blood biomarker among the two groups that can show if these markers (tau, UCHL1, and GFAP) are altered among the two groups which can add to the conclusion

the Participant section should include demographics of patients along with the exclusion or inclusion criteria for the study, ie chronic disease, smokers etc..

Author Response

Dear Reviewer 1

Thank you for reviewing our manuscript and providing suitable comments.

We have made the following corrections to the manuscript, as per your suggestions.

The revised parts are shown in red font in the text.

The title and the overstatement in abstract

We have added, “a pilot study for early detection of cognitive decline," to the title.

In the abstract, the overstatement of, “these results of blood flow were suggested to be used as biomarkers for cognitive decline,” was revised to read as follows: “The differences in cerebral blood flow under the pre-frailty conditions may be a clue to detecting cognitive decline earlier.”

About Ethics notation

We have added a section titled “2.1. Ethical Considerations” in the Materials and Methods section to address the ethical concerns noted.

About consideration of another biomarkers

Unfortunately, it was not in the scope of this study to measure other biomarkers. However, this is a very interesting point of view, and we will use it as a reference for future research.

Kind regards,

Kohei Maruya

Reviewer 2 Report

In this research paper, the authors investigate how different non-invasive biomarkers vary as mental cognition deteriorates towards dementia. The purpose is to identify "pre-frail" subjects before dementia sets in. They use chi-square, ANOVA and t-test to differentiate between the groups.

The paper is scientifically sound and presented clearly and concisely and thus is easy to follow. 

The results require further investigation and the methodology lacks novelty.

However, the direction of this work is important in the healthcare sector.

Author Response

Dear Reviewer 2

Thank you very much for reviewing our paper. We are also very grateful for your kind and encouraging words. We will use your comments as a reference for our future research.

We have revised the manuscript as per the suggestions of the other reviewers. The corrections have been made in red font.

Kind regards,

Kohei Maruya

Round 2

Reviewer 1 Report

answered my queres